# Simulation-Based Inference with Uncertainty Quantification using Generative Models in Quantum Chromodynamics

## Abstract

Generative and adversarial machine learning methods have been used for parameter inference of physical models from observed data in various works. However, many real-world problems of interest involve non-differentiable models, a context in which many approaches cease to be sufficient. An example of this can be found in quantum chromodynamics, where inferring quantum correlation functions from observed data is hindered by the problem's intrinsic non-differentiability and stochasticity. To overcome this, we present a framework based fundamentally on generative adversarial networks in which parameters are iteratively optimized to generate realistic samples. This framework is novel compared to related works in that it simultaneously circumvents non-differentiability, enables uncertainty quantification, and is free of assumptions on parameters. We demonstrate the utility of this framework in learning synthetic distributions and simulated quantum correlation functions.

## 1 Introduction

Parameter inference and its associated uncertainty quantification in scientific modeling is a cornerstone of science, with many standardized techniques and tools available that enable domain researchers to stress-test postulated models and theoretical frameworks against physical systems (Rastogi (2021); Gábor & Banga (2015); MacLeod (2020)). In recent years, generative modeling in machine learning (ML) has found applications, in this context, with potential capabilities that could surpass existing methods in their performance on high-dimensional, many-parameter models (Kutz (2023)). In particular, generative modeling has been applied in simulation-based inference in high-energy physics Chan et al. (2023); Andreassen & Nachman (2020); Cranmer et al. (2020).

One of the general challenges in using ML techniques is the inherent need to construct computational frameworks using differentiable programming to perform standard back-propagation for ML models. This requirement is particularly challenging for simulation-based inference, which involves multiple components that have undergone dedicated R&D over the years and are difficult—if not impossible—to rewrite with autodifferentiation capabilities.

An example of this occurs in nuclear particle physics, where researchers aim to reconstruct from observational data, using high-energy scattering experiments, the internal quark and gluon structures inside nucleons and nuclei. There are dedicated and exciting programs in this field, such as those at Jefferson Lab 12 GeV (McKeown (2011)), COMPASS at CERN (Abbon et al. (2007)), RHIC at BNL (Aschenauer et al. (2014)), and the planned Electron-Ion Collider (Khalek et al. (2022)), where the development of generative modeling for inference on end-to-end simulations could become critical.

In this work, we present a case study of using generative ML in simulation-based inference with uncertainty quantification in the context of hadron structure studies that bypasses the autodifferentiation requirements. We briefly discuss a simplified simulation pipeline that will serve as a test bed for our studies. Then, we formulate the inference problem in the context of generative modeling using Generative Adversarial Networks (GANs) and discuss our strategy to avoid issues with back-propagation. Our main contributions are as follows:

**Non-Differentiable Model Parameter Learning:** Our approach is able to accomplish parameter inference on underlying physical models for which analytic differentiation is either unavailable or prohibitively challenging. We demonstrate the ability to circumvent such non-differentiability and seamlessly integrate with arbitrary application code with stochastic sampling.

**Parameter Uncertainty Quantification:** The nature of our generative setup gives straightforward *uncertainty distributions* over the inferred parameters, effectively learning prior distributions over parameters from data. We emphasize the utility of this approach in cases where we *do not* have assumptions we can make on the priors (or wish to not impose any such biases), so being able to empirically form distributions over feasible parameters is crucial in interpretability when no other knowledge is available. We focus our attention on *epistemic* (or model-centric) uncertainty in this paper and leave aleatoric (or data-centric) uncertainty as an application-specific concern.

**Reduced Training Dynamics Complexity:** Similar approaches to this problem tend to require additional neural networks to address the inner non-differentiability. This is accomplished either through training of probabilistic surrogate event generators (which can add a second inner adversarial loop) or offline fitting of a differentiable surrogate physical model approximation. Both of these are high-dimensional and complex mappings with non-trivial training cost. In contrast, our approach only requires training of a single additional model, which is lightweight and learns a mapping simply from the parameter space to the discriminator output.

**Assumption-Free Inference:** We impose no assumptions on the parameter range or prior distributions. In phenomenological contexts where we may be simultaneously developing theory to fit to observed data, this is a desirable paradigm.

## 2 BACKGROUND

To illustrate the domain physics problem for our case study, we consider the so-called deep-inelastic scattering (DIS). This process is characterized, for example, by the scattering of a high-energy beam of electrons off a beam of protons. At distance scales of $10 \times 10^{-15}$ m, the highly energetic incoming electrons have a small wavelength that can penetrate deep inside the hadron and interact (or scatter off) with quarks or gluons –collectively known as *partons*–, which are the elementary constituents of protons. The scattered electron momenta are then recorded by detectors around the interaction region, and this information can be used to infer the longitudinal distribution of quarks and gluons inside the proton. Using the theory Quantum Chromodynamics (QCD), one can write schematically the Phase Space Density (PSD) of the outgoing electron on a proton target $(p)$ as

$$\rho_{(p)}(x, Q^2|\theta) = \sum_i \int_x^1 \frac{d\xi}{\xi} \mathcal{H}_i \left( \frac{x}{\xi}, Q^2 \right) f_{i/p}(\xi, Q^2|\theta) \,. \tag{1}$$

Here, $f_{i/p}$ is known as the *Parton Distribution Function* (PDF)[1], which represents the number density for finding a parton of flavor $i$ (up, down, strange, charm, bottom, gluon, and anti-quarks) inside the proton with a longitudinal momentum fraction $\xi$ between $\xi$ and $\xi + d\xi$. In contrast to the coefficients $\mathcal{H}$, which are calculable in perturbative QCD, the PDFs are not calculable from first principles and need to be inferred from data using a parametrization. We explicitly annotate the parametrization dependence of the PSD and PDFs with $\theta$. The quantities $x$ and $Q^2$ are defined in terms of the momentum variables entering the process: specifically, $Q^2 = -q^2 = -(l - l')^2$ and $x = Q^2/(2P \cdot q)$, with $l, l'$, and $P$ being the incoming and outgoing electron momenta and the proton momentum, respectively. Our goal in this work is to infer multiple PDFs $f_{i/p}$, which are functions parametrized by some $\theta$: thus, our goal becomes inferring the feasible values of $\theta$.

Traditionally, a numerical strategy known as *unfolding* is used to remove detector effects and backgrounds, thereby reconstructing the pure electron phase space density and carrying out the inference directly at the density level. However, this approach is subject to irreducible systematic uncertainties associated with unfolding algorithms, which require the use of external models since it is technically an inverse problem. An alternative approach is the aforementioned end-to-end simulation to infer PDFs, aiming to mitigate such irreducible uncertainties. The simulation pipeline can be written

---

[1]Whenever the abbreviation PDF is used in this paper, it is referring to the domain-specific term "Parton Distribution Function." When discussing probability distribution functions, we will do so explicitly.

schematically as

$$
\begin{aligned}
\theta &\to \rho_{(p)}(x, Q^2|\theta) \\
&\to (x, Q^2) \sim \rho_{(p)}(x, Q^2|\theta) \\
&\to \text{Detector simulator + backgrounds} \\
&\to \text{simulated } (x_{\text{sim.}}, Q^2_{\text{sim.}}) \ .
\end{aligned}
\tag{2}
$$

From an optimization point of view, the task is to construct a distance metric between the simulated samples[2] $(x_{\text{sim}}, Q^2_{\text{sim}})$ and the experimental samples, and use it to make updates on $\theta$. If the inference on the latter involves the use a generative algorithm, there is a requirement for autodifferentiation across all components in Eq. (2). For instance, for a GAN approach we have schematically

$$
\begin{aligned}
(x_{\text{exp.}}, Q^2_{\text{exp.}}) &\to D \to \text{Score} \to \text{D.Loss} \to \text{back.prop } (D) \\
z \to G \to \theta \to \text{Eq. (2)} \to (x_{\text{sim.}}, Q^2_{\text{sim.}}) &\to D \to \text{Score} \to \text{D.Loss} \to \text{back.prop } (D) \\
z \to G \to \theta \to \text{Eq. (2)} \to (x_{\text{sim.}}, Q^2_{\text{sim.}}) &\to D \to \text{Score} \to \text{G.Loss} \to \text{back.prop } (G)
\end{aligned}
\tag{3}
$$

Here $G$ and $D$ are the standard generator and discriminator respectively. The $G$ transforms a latent space $z$ to $\theta$ space which are passed to the simulation chain in Eq. (2) to produce simulated phase space samples. The latter are passed to the discriminator to assign a *score* and use it in loss to make updates on $G$. Concurrently, the discriminator is updated using the real experimental phase space samples $(x_{\text{exp.}}, Q^2_{\text{exp.}})$.

It should be noted that a vanilla GAN (Goodfellow et al. (2014)) cannot be used to implement Eq.(3) for several reasons. First, and most importantly, the simulation pipeline in Eq.(2) involves a detector simulator, and state-of-the-art simulators are not differentiable through numerical or autodifferentiation approaches (Allison et al. (2006; 2016); Agostinelli et al. (2003)). Second, the phase space samples need to be drawn from $\rho_{(p)}(x, Q^2|\theta)$. While there are proposed solutions for approximating gradients of samples with respect to the parameters of the corresponding probability distributions (see (Fu, sec. 4) and (Figurnov et al., sec. 5)), extension to high-dimensional random vectors or massive-parameter problems such as those found in state-of-the-art domain-specific simulations can be very expensive and difficult to integrate in common machine learning pipelines for such problems. Even if the latter issue could be addressed with a clever algorithmic procedure, the first problem is unlikely to be solved due to the complexity involved in detector simulators.

This work sits at the intersection of many similar proposed solutions for related problems in adversarial machine learning, non-differentiable parameter inference, and PDF fitting. The naive classical approach of directly searching the parameter space is infeasible in this application due to the assumption that prior parameter ranges and distributions are unknown, thus motivating the need for advanced parameter inference enabled by machine learning. We summarize the most closely related works to this paper and their key contributions and differences in Table 1. In particular, a surrogate event generator was used by Alghamdi et al. (2023) in order to construct a neural network which learns to generate simulated observable events from parameters. This involves a concurrent training setup with both inner and outer GANs all learning simultaneously. While there is strong success in the generation of high-quality synthetic data from parameters using this surrogate event generator, training requires large amounts of data, which may be costly to obtain. Moreover, many similar works in stochastic parameter inference constrain possible models entirely to differentiable ones: in such cases, non-differentiability is accommodated through offline fitting of a differentiable surrogate model to effectively replace the non-differentiable physical model (Rumbell et al. (2023)). This surrogate fitting can be a non-trivial computational task with possible nuance lost through the imposed approximation. Advanced methods such as Adversarial Variational Optimization (Louppe et al. (2019)) which directly tackle non-differentiable models do so through forced priors, which we wish to avoid in this case where parameter knowledge may be entirely unknown.

Moreover, many of these works do not explicitly address uncertainty quantification (UQ), which is crucial in ensuring the trustworthiness of extracted PDFs. Many well-known formulations of UQ in the more general literature of parameter inference generally impose Gaussian priors on the parameter

---

[2]We use the term "sample" to refer to a single event (1-D in this paper, without loss of generality). In the context of a toy distribution with probability distribution function given by $f(x; \theta)$, a sample would be $x \sim f(x; \theta)$. In QCD an event would be a single tuple $(x_{\text{sim}}, Q^2_{\text{sim}})$.

| Paper | UQ | # NNs | Prior-Free | $f'(x)$ | Training Cost |
|---|---|---|---|---|---|
| AVO Louppe et al. (2019) | ✓ | 1 | ✗ | ✓ | Medium |
| GAN-based $\theta$ estimation (Rumbell et al. (2023)) | ✓ | 3 | ✓ | ✗ | High |
| Inner/Outer GANs (Alghamdi et al. (2023)) | ✓ | 4 | ✓ | ✓ | High |
| GMMs for PDF Fitting (Yan et al. (2024)) | ✓ | 0 | ✗ | ✓ | High |
| **This Work** | ✓ | 3 | ✓ | ✓ | Medium |

Table 1: Summary of the most closely related approaches to solving similar inverse problems motivated by QCF extraction. The $f'(x)$ column refers to whether the work directly accommodates non-differentiability. If the work does off-line differentiable surrogate fitting to accommodate non-differentiability, we mark this as ✗ in this column.

distributions (Bui-Thanh et al. (2012); Lele (2020); Abdar et al. (2021)). We deliberately wish to *avoid* enforcing any kinds of Gaussian priors on the distributions in the design of this framework: such an assumption would prevent insights into possible skewness of the underlying distribution of generated parameters. As such, instead of using mean and standard deviation as measures of spread, we use median and percentile parameters and present interquartile ranges (IQR) of parameters as a proxy for epistemic uncertainty.

For our studies, we will simplify the problem by not considering the detector simulations nor backgrounds in Eq.(2). In the next section will discuss our strategy to implement our GAN parameter inference framework by supplementing the GAN architectures with differentiable ML surrogates.

## 3 METHODS

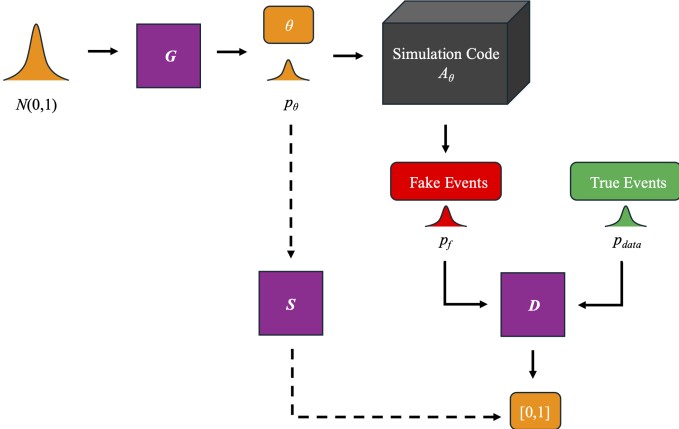

Figure 1: Schematic of network and feedforward propagation paths in our GAN-based approach with distributions generated at each stage of training. During training of the generator, the output is propagated along the dashed line.

To address the issue of infeasible automatic differentiability of the domain problem, we introduce a third network which we will refer as a *Score Prediction Network* (SPN), or $S$ in algorithmic contexts. This network aims to learn the score of parameters that the discriminator assigns to the corresponding phase space samples. The training samples for SPN are constructed directly using the simulation code in Eq.(2) at a given state of the discriminator. This SPN is trained as a sub-loop of discriminator training. Specifically, whenever $D$ weights are updated, we also update the $S$ weights to map correctly the transformation from parameters $\theta$ to the discriminator $D$. As a result,

---

**Algorithm 1:** Algorithm depicting the training process. We consider only a single sample drawn here and do not account for batches in this pseudocode. More details regarding our particular implementation of batching can be found in Section 3. $A_\theta$ is the density of an application-specific simulator.

---

**while** not converged **do**
    **for** $N_D$ *steps* **do**
        Sample $z \sim \mathcal{N}(0, \mathbf{I})$ and let $\theta = G(z)$;
        Sample $\mathbf{x} = (x_0, \ldots, x_{31}) \sim A_\theta$ and $\mathbf{x}_D = (x_0^{(D)}, \ldots, x_{31}^{(D)}) \sim p_{data}$;
        Take gradient descent steps on $\nabla_{\theta_D} \|D(\mathbf{x}) - \mathbf{1}]\|_2^2$ and $\nabla_{\theta_D} \|D(\mathbf{x}_D) - \mathbf{0}]\|_2^2$;
        **for** $N_S$ *steps* **do**
            Sample $\mathbf{x}_S = (x_0^{(S)}, \ldots, x_{31}^{(S)}) \sim A_\theta$;
            Take gradient descent step on $\nabla_{\theta_S} \|D(\mathbf{x}_S) - S(\theta)\|_2^2$;
        **end**
    **end**
    **for** $N_G$ *steps* **do**
        Sample $\mathbf{x}_G = (x_0^{(G)}, \ldots, x_{31}^{(G)}) \sim A_\theta$;
        Take gradient descent step on $\nabla_{\theta_G} \|D(\mathbf{x}_G) - \mathbf{1}\|_2^2$;
    **end**
**end**

---

we modify the GAN setup in Eq.(3) to

$$
\begin{aligned}
(x_{\text{exp.}}, Q_{\text{exp.}}^2) &\rightarrow D \rightarrow \text{Score} \rightarrow \text{D.Loss} \rightarrow \text{back.prop} \ (D) \\
z \rightarrow G \rightarrow \theta \rightarrow \text{Eq. (2)} \rightarrow (x_{\text{sim.}}, Q_{\text{sim.}}^2) &\rightarrow D \rightarrow \text{Score} \rightarrow \text{D.Loss} \rightarrow \text{back.prop} \ (D) \\
z \rightarrow G \rightarrow \theta \rightarrow \text{Eq. (2)} \rightarrow (x_{\text{sim.}}, Q_{\text{sim.}}^2) &\rightarrow D \rightarrow \text{Score} \rightarrow \text{S.Loss} \rightarrow \text{back.prop} \ (S) \\
z \rightarrow G \rightarrow \theta \rightarrow S &\rightarrow \text{Score} \rightarrow \text{Loss} \rightarrow \text{back.prop} \ (G) \ .
\end{aligned}
\tag{4}
$$

Our proposed GAN+SPN only requires lightweight models with cheap storage costs and deals with optimizing a simpler, lower-dimensional, and more feasible subproblem: mapping from parameter space to the discriminator output. Our overall training process is outlined in Figure 1 and described more explicitly in Algorithm 1. For notational simplicity, we may write $D(x; \theta_D)$ as simply $D(x)$ or $D$ and so on for other models, the *network* parameters $\theta_D$ (not the simulator parameters) being assumed in the model definition.

The discriminator learns on minibatches of 32 events drawn from the distribution parametrized by a single generator sample. The true data is shuffled and drawn in minibatches of the same size without replacement; we do not use any bootstrapping in this work. Training is terminated after 1000 generator updates. In this paper, our true dataset is constructed from synthetic events in which the parameters of the simulators are manually defined. This is advantageous for methodological development purposes: if one knows the true parameters, one can determine whether this approach converges to the parameters with which the data was generated. In integration with experimental studies, of course, the true dataset would consist of real observed samples.

## 4 THEORETICAL ANALYSIS

We now briefly consider the theoretical results produced by Goodfellow et al. in the seminal GAN paper. We show that under specific conditions of the (fixed) physics and surrogate models, many of their theoretical results hold for GAN+SPN; that is, that with large enough models, we may always find an optimal generator or discriminator given that the other one is fixed. We denote the distribution $p_g$ as the distribution of generated parameters, $p_f$ as the distribution of generated events for some fixed $\theta$, and $p_{data}$ as the distribution of true events. These are also labeled in Figure 1.

Let $\theta \in \mathbb{R}^d$ parametrize some application-specific simulator whose density is given by $A_\theta$. If we consider the generator in their proofs to refer to the generator $G$ and application simulator $A_\theta$ applied

in sequence as composite functions, then the proofs of Proposition 1 and Theorem 1 in Goodfellow et al. (2014) hold. Specifically, Proposition 1 states that given a fixed generator $G$, we may always find an optimal discriminator $D_G^*$. This is done via optimization of the training criterion

$$V(G, D) := \mathbb{E}_{x \sim p_{data}} \left[ \log D_G^*(x) \right] + \mathbb{E}_{x \sim p_f} \left[ \log \left( 1 - D_G^*(x) \right) \right]. \tag{5}$$

In their work, $V(G, D)$ is sometimes written equivalently as $U(p_f, D)$ to emphasize the training criterion being a function of the generated distribution. Theorem 1 states that we can only attain the global minimum of the training criterion if $p_g = p_{data}$. We now present an extension of Proposition 2 in Goodfellow et al. (2014) to the case of a non-differentiable sampling component. We introduce one new assumption: we require that $A_\theta$ is *well-conditioned* in the sense that small perturbations to $\theta$ result in proportionally small perturbations to the shape of the distribution of $A_\theta$. Precisely, there must exist sufficiently small $\kappa > 0$ such that for all possible $\theta$ in the parameter space, if we are given $\theta'$ such that $\|\theta' - \theta\|_2^2 < \delta$ for all $\delta > 0$, we have

$$\|A_{\theta'} - A_\theta\|_{JSD}^2 < \|\theta' - \theta\|_2^2 \kappa \tag{6}$$

where $\| \cdot \|_{JSD}$ refers to the Jensen-Shannon divergence (Fuglede & Topsoe, 2004). What this assumption tells us is simply that updates to $\theta$ are proportional to updates to the induced distributions $p_f$ from $A_{\theta'}$ in the aforementioned distance metrics.

We also make the possibly strong assumption in Proposition 1 that $S$ perfectly approximates the prediction of the discriminator given a set of predicted parameters. Based on our experiments, we suspect that this assumption is a gross overestimation and that it may be relaxed to some $\varepsilon$ tolerance. We also suspect that there is a more precise statement to be made about how large $\kappa$ may be in order to ensure proportional updates of $p_f$ with respect to $p_t$. We leave this for future work.

**Proposition 1.** Suppose we have large enough $G$ and $D$, optimal $D$ at each iteration and that $S$ is a perfect mapping from $G$ output to $D$ output. Then if $A_\theta$ is well-conditioned in the sense of Eq. (6), and updates to $p_f$ made so as to improve the training criterion in Eq. (5), then $p_f$ converges to $p_t$.

*Proof.* As in Theorem 1 in Goodfellow et al. (2014) and thus here, $\sup_D U(p_f, D)$ is convex in $p_t$. Then, there exists a global minimum to the above value function so long as we make sufficiently small updates to $p_f$. The only way we may do this is through proportionally sufficiently small updates to $p_g$ through an update of $G$, which is enabled by assumption. $\square$

Naturally, one may be concerned about whether we may reliably use the surrogate prediction model for training the generator. We provide the following proof to demonstrate that with appropriately designed generator and trained surrogate prediction models, we may comfortably use the surrogate prediction output in the absence of differentiability from discriminator output. This result is independent of any assumptions on the conditioning of $A_\theta$.

**Theorem 1** (Surrogate Accuracy). Fix $\theta \in \mathbb{R}^d$, where $d$ is the dimension of $\theta$. Assume the space of permissible parameters is $\mathbb{R}^d$. Define the space $\mathbb{S}(\theta)$ to be the event space of $A_\theta$. Let $G : \mathbb{R}^n \to \mathbb{R}^d$ and $D : \mathbb{S}(\theta) \to \mathbb{R}$. Let $S : \mathbb{R}^d \to \mathbb{R}$. Finally, suppose that $G$ is surjective onto $\mathbb{R}^d$. Suppose $\exists \varepsilon > 0$ such that $\forall s \in \mathbb{S}(\theta)$

$$\|D(s) - S(\theta)\|_2^2 < \epsilon.$$

Then, there exists $z$ such that

$$P_{s \sim A_{G(z)}}(\|D(s) - S(G(z))\| < \epsilon) = 1$$

*Proof.* By assumption, there exists a subset of the event space such that $\|D(s) - S(\theta)\| < \varepsilon$. We assume that the complement of this set has probability zero. Consider arbitrary $s \sim A_\theta$ from this subset. We know that $G$ is surjective onto $\mathbb{R}^d$, therefore it is able to generate the given $\theta$: precisely, there exists $z \in \mathbb{R}^n$ such that $\theta = G(z)$. Then, at this $z$, we have

$$D(s \sim A_\theta) = D\left(s \sim A_{G(z)}\right) \implies \left\|D\left(s \sim A_{G(z)}\right) - S(G(z))\right\|_2^2 < \epsilon.$$

$\square$

This simply tells us that if we can train $S$ to sufficiently approximate the output of $D$, we may confidently use it as a surrogate mapping directly from parameters to discriminator output. This is contingent upon ensuring that $G$ can, indeed, map to the entire parameter space $\mathbb{R}^d$. We may do this by ensuring $G$ is the composition of surjective functions. In practice, as we use a simple multi-layer perceptron (MLP) as the generator with no parameter clipping, this is guaranteed.

We also note that the generator has the capacity to approximate the parameter prior distributions with arbitrary desired Wasserstein distance closeness, according to Yang et al. (2022). This is useful from a UQ standpoint, however, as in the original GAN paper, this is again under idealized circumstances. The main inhibitor we encounter in our work is data availability and model well-posedness: a true dataset which does not sufficiently cover the possible event space as well as physical models for which multiple distinct parameters can map to the same distribution could both lead to incorrect parameters (and thus empirically formed parameter prior distributions). We emphasize that in the PDF application, however, parameter inference can be thought of as a means to an end, as our true priority is extracting the PDFs, for which parametrizations are somewhat arbitrary and can even be replaced by a neural network entirely. The choice of a thoughtfully parametrized PDF is useful for interpretability and uncertainty quantification, and our data-driven approach enables these without any other imposed assumptions.

## 5 RESULTS

All experiments are done on GPU nodes on the Perlmutter supercomputer: 4x NVIDIA A100 GPUs (40GB) per node. We do not provide scaling studies here with respect to numbers of GPUs involved. In future work, to more rapidly accelerate training in an online setting, it may be useful to investigate high-performance variants of GANs to ensure efficient and rapid parameter estimation. The SAGIPS paper (Lersch et al. (2024)) provides some insight into how this scaling could be accomplished.

This section presents results on GAN+SPN for two distributions: first, learning the rate parameter of a Poisson distribution, and second, learning the defining parameters of simple PDFs from events sampled from phase space densities. For visual examples, we present the generated Poisson events in histogram form in Figure 2 and the reconstructed PDFs in Figures 5 and 4. For ablation studies investigating the sensitivity of our results on the PDF problem to training-related hyperparameters, we refer the reader to the appendix.

For the Poisson distribution results, we present results compared with AVO. We clarify that we present this comparison to AVO purely to contextualize this work among other Bayesian inference approaches towards non-differentiable parameter estimation. However, we emphasize that the allure of this particular method is that, unlike AVO, at no point do we enforce any assumptions on the priors of the parameters, while known or estimated priors on the parameters are assumed and harnessed in their variational optimization approach. This difference is crucial when we consider the specific physics application problem, where being able to learn possible parameters without additional inductive bias of parameter priors is a crucial draw of this framework.

### 5.1 POISSON DISTRIBUTION

We use a Poisson distribution with defining parameter $\lambda > 0$ as an artificial non-differentiable simulator. Our framework is used to learn 15 distinct lambdas between 0 and 4 as in the first experiment in Louppe et al. (2019). We make the note that the Poisson distribution is desirable as proof-of-concept example for multiple reasons: it is 1) discrete and thus immediately non-differentiable and 2) uniquely defined by a single parameter. For the AVO experiments, we use the default parameters in their implementation and use the same proposal distribution for each lambda experiment. Results for this comparison are presented in Figure 2. We also present an illustrative example with parameter uncertainties formed over summary statistics over 32 sampled parameters at each generator training step. In Louppe et al. (2019), the authors make the realistic assumption that simulated sampling may be costly and thus enforce the notion of a simulation budget, limiting the total number of samples which may be drawn during optimization. We maintain this assumption and terminate training once we reach the simulation budget of 160,000 samples.

In the rightmost plot in Figure 2, we see that GAN+SPN vastly outperforms AVO on parameter accuracy. In Louppe et al. (2019), the authors further benchmark AVO against state-of-the-art ap-

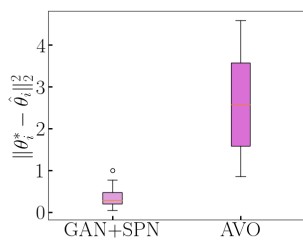 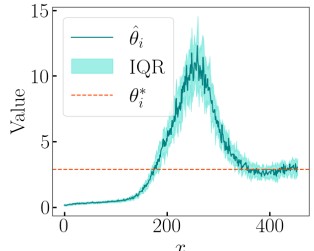 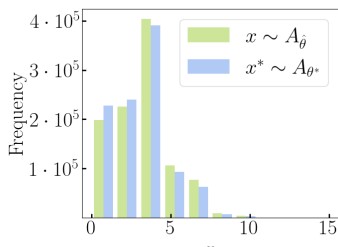

Figure 2: **Left**: average error of recovered true parameters between true and generated distributions between GAN+SPN and AVO. **Center**: parameter convergence for a single $\lambda$ over training and corresponding histogram of generated events. **Right**: histogrammed generated events using parameters from GAN+SPN for the same lambda as the center figure.

proaches ABC-SMC (Toni (2011)) (important implementation of Approximate Bayesian Computation (Sunnåker et al. (2013))) and BOLFI (Gutmann et al. (2016)) (similar likelihood-free approach based on optimizing over summary statistics). We use the implementation of AVO publicly available online[3] and construct a 3-layer MLP with 600 nodes at each hidden layer as the "critic" network (similar to a discriminator), but are unable to duplicate their results using the default parameters they provide online. While we cannot replicate the results with average error that they provide in their paper, AVO was demonstrated to outperform both ABC-SMC and BOLFI on both the single-dimensional Poisson example and other multidimensional distributions on the metric of parameter error. A brief study into optimizing the AVO parameters to obtain its reported high accuracy was done, and while further optimization of both GAN+SPN and AVO is required, this may possibly suggest potential of our approach compared to leading methods in prior-informed parameter inference.

The results in Figure 2 demonstrate that GAN+SPN outperforms default AVO settings on all metrics with no priors imposed on the estimated parameters. In particular, we observe that with a simulation budget of just 160,000 (about $\sim 10\%$ of the available dataset), the inferred parameters converge to the true $\lambda$ a little over halfway through using up this budget. The final average parameters generate events which are nearly indistinguishable in aggregate as a histogram from the generated data.

## 5.2 PROXY PDF PROBLEM

Instead of working directly with the full QCD PDF problem discussed in the introduction, we perform our analysis on a simplified version of the problem; we remove many aspects of real physics simulations while retaining the essential features necessary for simulation-based inference solutions via GAN+SPN. Specifically, we focus on proton and neutron PSD, so that Eq. (1) becomes

$$\begin{aligned} \rho_p(x|\theta) &= 4u(x|\theta) + d(x|\theta) \\ \rho_n(x|\theta) &= u(x|\theta) + 4d(x|\theta) \ . \end{aligned} \tag{7}$$

Here $u, d$ are the up and down quark PDFs parametrized as

$$\begin{aligned} u(x|\theta) &= N_u x^{a_u}(1-x)^{b_u} \\ d(x|\theta) &= N_d x^{a_d}(1-x)^{b_d} \end{aligned} \tag{8}$$

To simplify the problem without losing generality, we fix the normalization parameters to $N_u = 2$ and $N_d = 1$, respectively, to mimic the net valence quark content of physical nucleons, leaving four free parameters in the problem: $\theta = [a_u, b_u, a_d, b_d]$.

Here, the events are PSD samples in $x$ drawn independently from $\rho_n$ and $\rho_p$ and evaluated by two independent discriminators, $D_n$ and $D_p$, which share the same architecture. A single generator is used to produce the parameters $\theta$, and a single SPN is employed to map from $\theta$ to the average combined prediction of $D_n$ and $D_p$ based on the given events.

---

[3]`https://github.com/neychev/adversarial_variational_optimization/blob/master/first_experiment.ipynb`

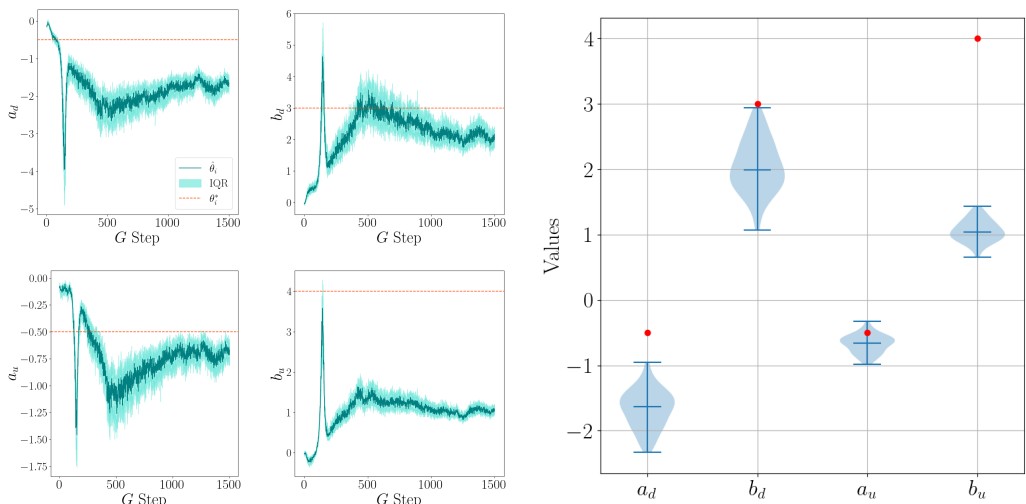

Figure 3: **Left:** Parameter convergence over generator updates for each of the parameters. **Right:** Violin plot of each of the four parameters, obtained through 100 samples of the trained $G$. The red dots represent the true parameters. Here, the total dataset size is 102,400.

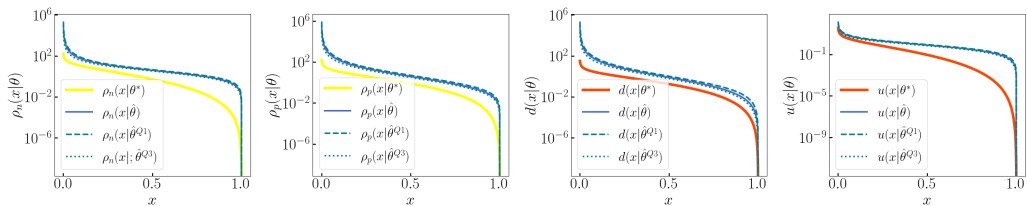

Figure 4: **Left:** PSDs $\rho_p(x|\theta)$ and $\rho_n(x|\theta)$. **Right:** reconstructed PDFs $u(x|\theta)$ and $d(x|\theta)$.

It should be noted that these parameters are correlated due to the model's parametric form; the $a$-type parameters control the small-$x$ behavior, while the $b$-type parameters govern the large-$x$ behavior, and in the intermediate region, both parameters contribute equally. In our analysis, we generated samples in the region $0 < x < 1$ so that the parameters do correlate strongly leading to potential biases in the inference. However, we are ultimately interested in the $u$ and $d$ quark PDFs rather than their parameters, and therefore, our metric of success in producing the ground truth is evaluated directly in the space of PDFs as well as the proton and neutron PSDs.

## 5.3 UNCERTAINTY QUANTIFICATION IN PDF FITTING

Our analysis of epistemic UQ in our method should be categorized in two closely-related ways. First, we are interested in the **distribution of the generated parameters.** We wish to show that as data availability increases, our certainty (inferred through the tightness of the empirical generated parameter distribution) increases. We model this in Figure 3 through modeling the final distributions of parameters. The violin plot of parameters clearly justifies our choice to use percentiles as opposed to Gaussian means and variances, as there is a clear skewness in parameters $b_u$ and $b_d$.

Second, we are interested in the **uncertainty of the reconstructed PDFs**; i.e. how uncertainty in the parameters propagates to the reconstructed PDFs (and simulated data). For the proxy example, this refers to the functions $u(x; \theta)$ and $d(x; \theta)$. We consider this aspect in two ways. In Figure 4, we consider parameter distributions independently to see how individual parameter uncertainty propagates to the functions, which gives results which closely approximate the results of the functions. In the reconstructed PDFs, we see that not only do $u(x; \hat{\theta}_u)$ and $d(x; \hat{\theta}_d)$ closely approximate the true PDFs, but that the true PDFs and 1-D densities overall lie within those functions parametrized by

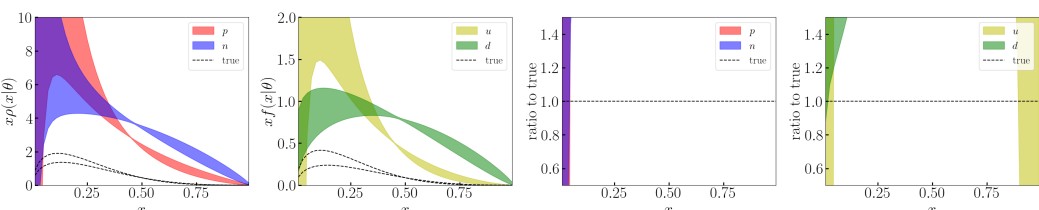

Figure 5: Recovered PDFs and their ratio to the true functions.

less certain estimates. This demonstrates that our method converges to a distribution of parameters which give us strong reconstructed PDFs. However, when we consider each of the generator outputs as parametrizing PDFs separately, we form a distribution of functions, which are described in Figure 5. Here, the reconstructed functions and their uncertainties are much further from the desired true functions. This could be a result of our implementation of minibatching. We leave this as our top priority in future work.

## 6 CONCLUSIONS AND FUTURE WORK

We present an adversarial ML framework which is able to infer parameters with respective uncertainty distributions from observed data. Our implementation only requires only one adversarial loop as opposed to other similar approaches and achieves differentiable training of the generator through a cheap, learnable surrogate mapping from the generated parameters to discriminator classification, eliminating the need for any offline preparatory work such as the potentially costly computation of a differentiable surrogate approximation of a physics model.

Other GAN variants, such as Wasserstein GAN (Arjovsky et al. (2017)) and Hinge (Lim & Ye (2017)) methods, have been developed to avoid some of the instability pitfalls often encountered with GANs. A brief investigation into replacing our standard GAN implementation with these approaches were done with as-of-yet inconclusive results. Future work may involve a more rigorous study into GAN modifications to accelerate convergence and improve stability of learned parameters.

We make the brief note that in the distributions that we examine, there is a so-called *unique* solution to the training objective (in an extremely idealized setting with infinite data availability); that is, that there is only a single set of parameters which uniquely define the underlying probability distribution functions. This may not be the case in more complex PDF parametrizations, where multiple distinct parameters could define identical observable event distributions. To combat this possible issue, our GAN+SPN framework can be easily extended to an *ensemble* setting in which multiple differently-initialized networks are trained on the data, enabling independent exploration of the parameter space. This would give us not only "intra"-model uncertainty estimates as we have in this work, but further uncertainty estimates of parameters through aggregation of multiple generator outputs. We briefly experimented with this approach but omit this in our quantitative analysis since the uniqueness of the solutions in the examples we present reduces the need for such ensemble learning.

The ultimate goal is to integrate this work into more complex state-of-the-art PDF fitting pipelines. This integration can give rise to further challenges, such as increased event dimensionality and more complex simulators with detector and background effects. While we anticipate networks with greater complexity and improvements to ensure adversarial training stability will be needed to accommodate these challenges, the uncertainty distributions constructed through this framework will enable an interpretable search of the possible parameter space in these large-scale problems.

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

APPENDIX

## A   MORE DETAILED LITERATURE REVIEW

**Machine Learning Techniques in QCF/PDF analysis**. A variety of works have investigated integrating machine learning into the QCF analysis pipeline in other ways. In Liu et al. (2022), neural networks are used to accelerate parameter space searching in PDF uncertainty calculations. Some works have proposed not developing analytical formulae to describe PDF behavior altogether and instead parametrizing PDFs by neural networks. Some examples include Bacchetta et al. (2022), Rojo (2006), and Dutrieux et al. (2022). A major open-source collection of work in using neural networks to directly and flexibly extract PDFs can be found in the NNPDF Collaboration, which encompasses some of the following highly useful papers in this area spanning over two decades: Forte et al. (2002); Del Debbio et al. (2007); Ball et al. (2009; 2021); Del Debbio et al. (2022).

**Machine Learning-Based Surrogate Models for PDFs: why fit to a parametrized model?** In Hunt-Smith et al. (2022), the authors show that on a small example, entirely neural network-based PDF extraction can aggravate uncertainty issues. Mitigating this uncertainty is an active area of research among NNPDF works (Ball et al. (2022; 2024)). Moreover, there is already an expansive amount of literature and well-developed mathematics for determining functions forms of PDFs (Novikov et al. (2020); Courtoy (2024)), as well as existing collaborations such as the Jefferson Lab Angular Momentum (JAM) Collaboration, from which we derive the code used for sampling

**Mode collapse in GANs.** Mode collapse is a frequently-observed and well-studied phenomenon in empirical GAN training dynamics (Zhang et al. (2018); Bhagyashree et al. (2020); Salimans et al. (2016)). We frequently encountered this in our own experiments, but were able to successfully employ a wide variety of known GAN training techniques to avoid this pitfall. In particular, we train the discriminator multiple times for each generator update, as is done in various prominent works such as Gulrajani et al. (2017). A training schedule of discriminator parameter updates at a rate of around 5 times more than generator parameter updates is considered common, and is thus the choice that we employ by default in this work. A study of the sensitivity of resulting metrics with respect to this rate can be found later in the appendix.

**Statistical divergence loss functions.** We use the standard GAN implementation in this work, which empirically accomplishes our needs for this application. However, other objective functions derived from minimizing statistical divergences between output and target distributions have become highly relied-upon, particularly in combating some of the instability issues mentioned earlier and commonly observed in other works. One of the most popular alternatives is Wasserstein GAN (WGAN) (Arjovsky et al. (2017)), which minimizes the Wasserstein metric as a proxy for the Earth Mover Distance (Rubner et al. (1998)). The $f$-GAN paper (Nowozin et al. (2016)) provides a detailed investigation of GANs using various statistical $f$-divergences. For additional details on GAN training, we recommend Mescheder et al. (2018). In future work, changing the objectives for the generator and discriminator networks in this paper can be handled by simple drop-in replacement and may be worth further investigation for improved stability. We briefly investigated eliminating a discriminator (and thus also surrogate) model altogether and instead training to minimize statistical divergences (such as Kullback-Liebler (Kullback & Leibler (1951)) and Sinkhorn (Genevay et al. (2018)), among others) between generated distributions and target data, but ultimately moved away from this due to its lack of differentiability. However, it may be useful to return to this in future work and again use a surrogate model to train a mapping from generated data distributions to a statistical divergence loss or score.

## B    IMPLEMENTATION DETAILS

We present some more details of our training implementation in this section.

Architecture and training details for the Poisson example are provided in Tables 2 and 3. These details for the QCF example are outlined in Tables 4 and 5.

| Model | # Layers | Widths | Activations | Dropout | # Parameters |
|:---:|:---:|:---:|:---:|:---:|:---:|
| $G$ | 2 | (16,6) | LeakyReLU, ReLU final | No | 817 |
| $D$ | 2 | (16,8) | ReLU, Sigmoid final | Yes | 177 |
| $S$ | 2 | (32,16) | ReLU, Sigmoid final | Yes | 609 |

Table 2: Architecture details used for the Poisson distribution example. When dropout is applied, it is done with probability 0.5 after ReLU activations.

| $D$ **LR** | $G$ **LR** | $S$ **LR** | **# $D$ steps** | **# $G$ steps** | **Noise Dimension** |
|:---:|:---:|:---:|:---:|:---:|:---:|
| 0.0001 | 0.0001 | 0.001 | 8 | 3 | 32 |

Table 3: Training configurations for the Poisson distribution example. The acronym LR stands for "learning rate." We use the PyTorch (Paszke et al. (2019)) implementation of the Adam optimizer (Kingma & Ba (2014)).

| Model | # Layers | Widths | Activations | Dropout | # Parameters |
|:---:|:---:|:---:|:---:|:---:|:---:|
| $G$ | 3 | (64,32,16) | LeakyReLU | Yes | 4788 |
| $D_n, D_p$ | 3 | (16,8,4) | LeakyReLU, Sigmoid final | Yes | 209 |
| $S$ | 2 | (64,32) | ReLU, Sigmoid final | Yes | 2433 |

Table 4: Architecture details for the PDF example.

| $D_n, D_p$ **LR** | $G$ **LR** | $S$ **LR** | **# $D$ steps** | **# $G$ steps** | **Noise Dimension** |
|:---:|:---:|:---:|:---:|:---:|:---:|
| 0.001 | 0.001 | 0.0001 | 8 | 3 | 32 |

Table 5: Training configurations for the PDF example.

**Output Range Restriction.** In the Poisson distribution, we impose a positivity constraint on the learned parameter through a ReLU activation at the end of the generator. This is the only parameter restriction we enforce.

**Minibatching.** We employ minibatching, a common technique in neural network training. In a single experiment, we set a fixed global minibatch size and use it in slightly different ways when training each of the three networks. To improve understanding, we describe our implementation of minibatching in depth below.

- $G$: With minibatching, a Gaussian noise tensor $z_{mb} \in \mathbb{R}^{minibatch\ size \times noise\ dimension}$ is given as input to the generator, which generates *minibatch size* number of parameters, which we store as a tensor $\theta_{mb} \in \mathbb{R}^{minibatch\ size \times d}$. Individually, each of the parameter estimates $\theta_{mb}(i,:)$ for $i$ in $(0, minibatch\ size)$ is used to parametrize a PDF and obtain a **single** event sampled from $A[x; \theta_{mb}(i,:)]$. This sampling process is distributed across multiple parallel processes for efficiency.

- $D$: From the generator and application code, we now have a tensor of sample events $(x, Q^2)$ aggregated and stored in $s \in \mathbb{R}^{2 \times minibatch\ size}$. The disciminator gives us predictions $D(s) \in \mathbb{R}^{1 \times minibatch\ size}$.

- $S$: The surrogate network learns to map from $\theta_{mb} \in \mathbb{R}^{minibatch\ size \times d}$ to $D(s) \in \mathbb{R}^{1 \times minibatch\ size}$.

## C    ABLATION STUDIES

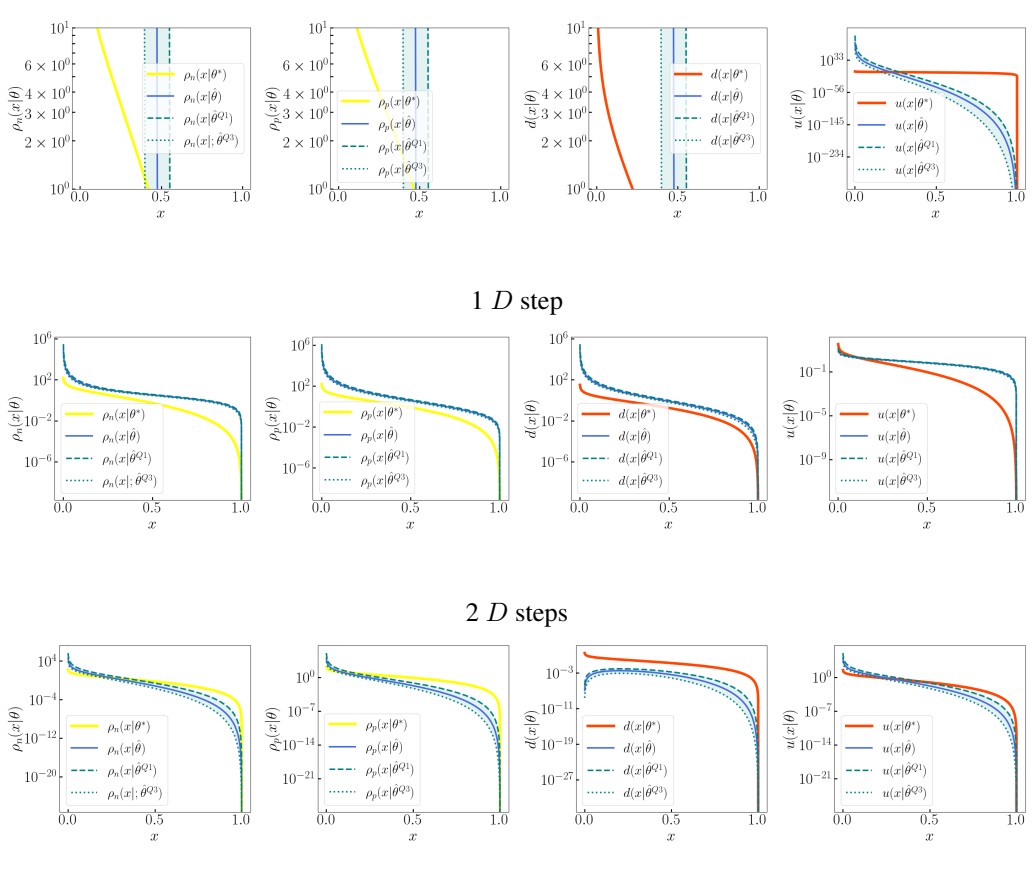

1 $D$ step

2 $D$ steps

5 $D$ steps

Figure 6: Sensitivity of PDFs with respect to the number of discriminator updates per generator update. As is expected, just a single update leads to too much instability and results in completely incorrect extracted functions. This improves with more $D$ steps, although $u(x; \theta)$ suffers at 5 $D$ steps.

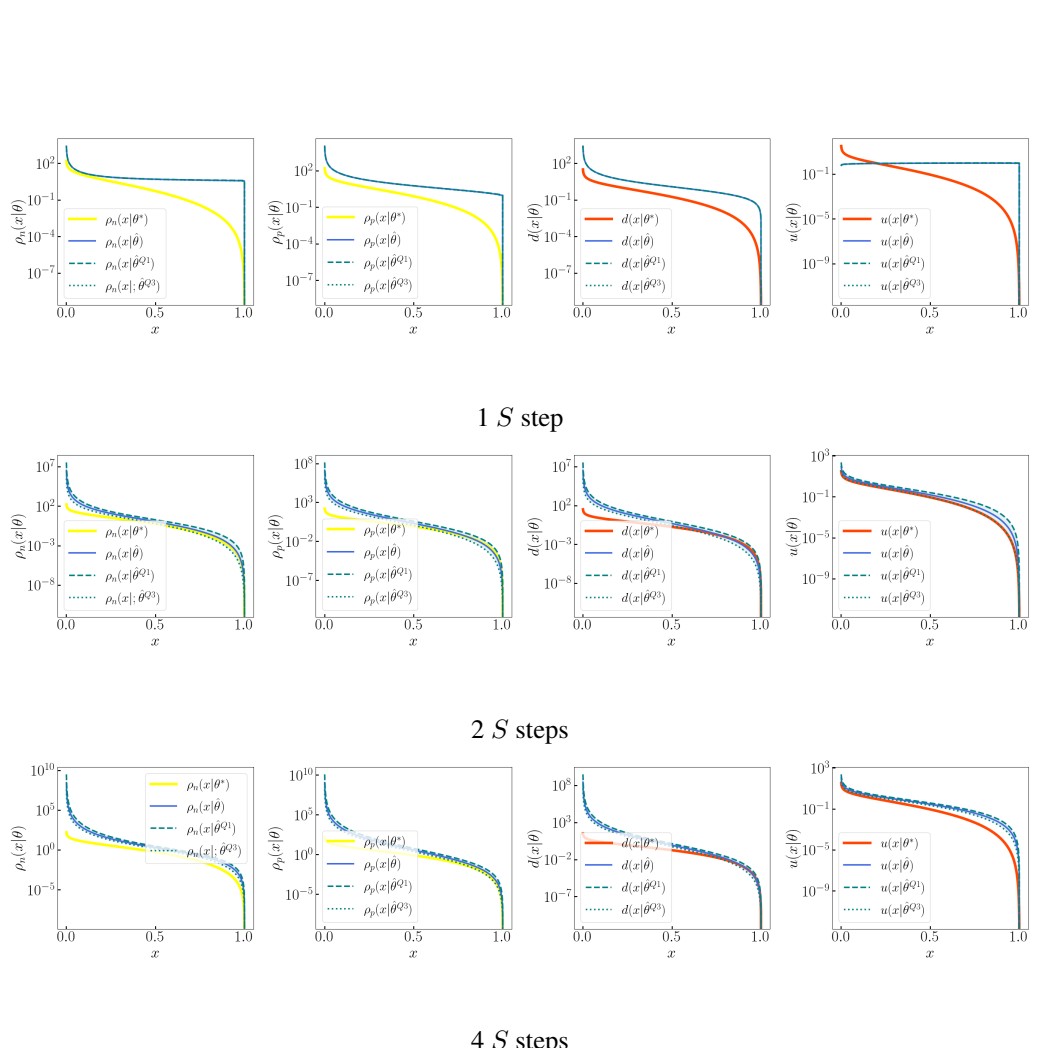

Figure 7: Sensitivity of PDFs with respect to the number of SPN updates per discriminator update. As expected, fewer SPN updates allow for error between SPN outputs and $D$ predictions to propagate back towards the generator updates, leading to poor parameters and PDF fits. We choose to limit the number of updates to $S$ at each discriminator update to avoid overfitting issues during training. We also aim to combat this issue through dropout in the SPN layers.

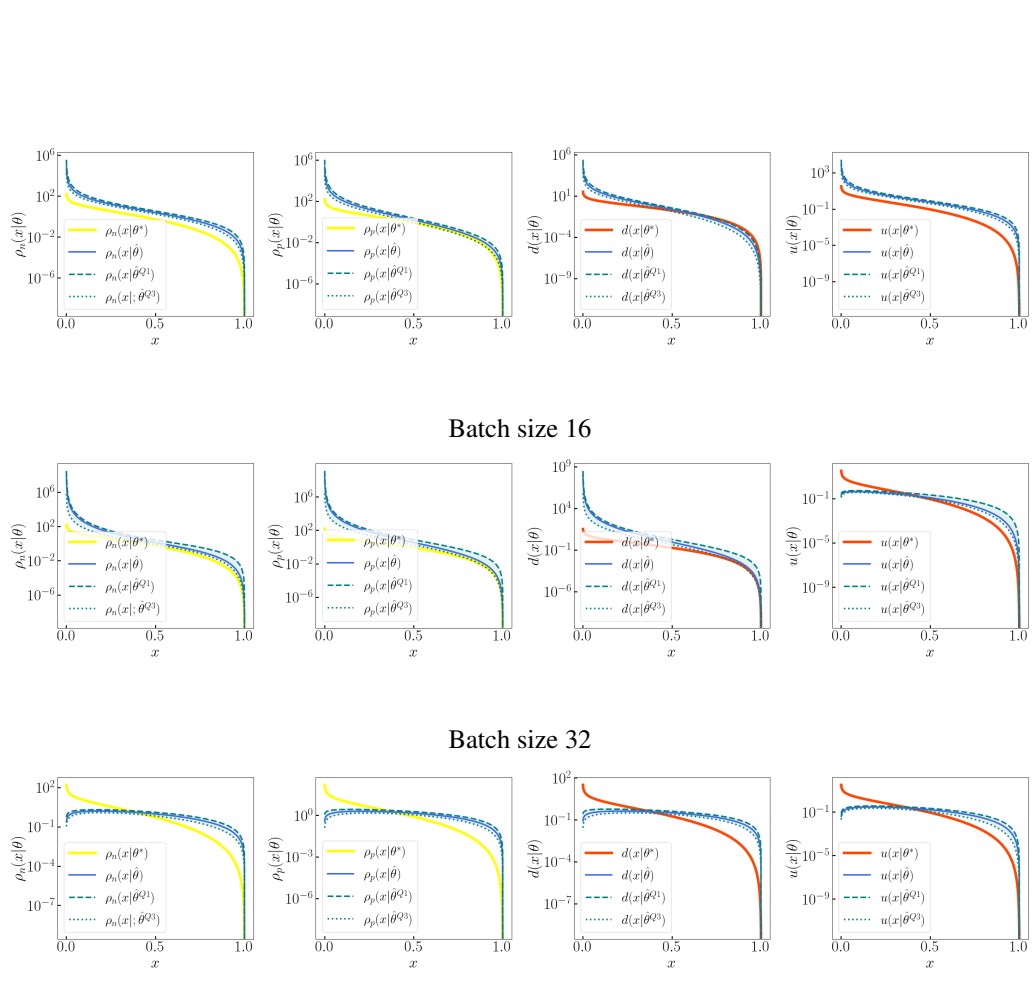

Batch size 16

Batch size 32

Batch size 128

Figure 8: Sensitivity of PDFs with respect to the batch size. We observe improved PDFs at relatively lower batch sizes (with the best PDFs extracted at batch size 64, which we present in the main results of this paper). Larger batch sizes may dilute effects of sampled events and lead to subpar PDFs, as we observe for the PDFs constructed with a batch size of 128 events.

