# OpenReview forum: "Simulation-Based Inference with Uncertainty Quantification using Generative Models in Quantum Chromodynamics"
_ICLR.cc/2025/Conference — ICLR 2025 Conference Withdrawn Submission_

### Official Review · Reviewer_aeQV · 2024-10-29

**Soundness:** 2
**Presentation:** 1
**Contribution:** 2
**Rating:** 3
**Confidence:** 2

**Summary:**

The authors propose a new method for estimating the parameters of quantum chromodynamics simulation. The resulting method allows uncertainty quantification of the parameters, does not require a differentiable simulator, and is based on generative adversarial networks.

**Strengths:**

The paper contains detailed explanations of the simulator and it seems that machine learning could be a potentially impactful method for this problem.

**Weaknesses:**

(1) Lack of relationship with simulation-based inference methods

I am confused about the relationship of the proposed method with other methods from the field of simulation-based inference (Cranmer et al 2020). The paper motivates their method by the fact that many simulators cannot (easily) be written in a differentiable programming language. However, simulation-based inference methods do not require gradients through the simulator. What, then, is the gap that this paper is filling?

(2) No discussion of poor calibration

In figure 3, the resulting distribution seems to be poorly calibrated (as the true parameters lie outside of the estimated distribution). Yet, there is no discussion of this behavior.

(3) Unclear writing

Many parts of the paper are written in an unclear way, leaving the reader to guess what exactly is meant. E.g.

L483 `which gives results which closely approximate the results of the functions`. What are `results of a function`? Function values?

(4) Poor results left for future work

In L500, the authors write that their method performs surprisingly poorly but that they will leave this to future work. Why not fix it now? It seems like an immediate thing that _should_ work and there is no reason to leave this for future work.

**Questions:**

Many parts of the paper are unclear to me. Formally (and with equations), what exact problem is the paper trying to address? Bayesian inference? Source distribution estimation? I think it is the latter, but I am really unsure. If it is the latter, how do the authors enforce maximum-entropy sources?

---

### Official Review · Reviewer_QGp3 · 2024-11-01

**Soundness:** 2
**Presentation:** 2
**Contribution:** 2
**Rating:** 5
**Confidence:** 4

**Summary:**

This paper propose a new method for inference without imposing the condition that the underlying model is differentiable. After explaining their model, the authors demonstrate a use case in quantum chromodynamics, with good results.

**Strengths:**

The study is well-motivated: a new approach for inference without the many assumptions in previous methods is pressingly needed, particularly for many applications in physics (where standard assumptions are violated). The authors claim that their proposed method (GAN+SPN) serves this purpose, as it does not need the underlying model to be differentiable and there are no assumptions on prior distributions. In addition, under their method GAN+SPN, they are able to reproduce many of the theoretical results in the original GAN paper.

**Weaknesses:**

The description of their method GAN+SPN is vague. While it is highlighted many times in the paper that GAN+SPN does not require the underlying model to be differentiable, it is unclear what the underlying model is and where differentiability (of what) is relaxed. The main explanations are made within one single physical example, which many of the conference participants may not understand.

On the theoretical side, most results simply follow from the same arguments in the original GAN paper (by Goodfellow et al.) or by imposing sufficiently strong conditions. The imposed conditions are quite strong (once they are assumed, the results almost follow immediately with minimal proofs needed) and it is unclear how they can be verified in practice. A stronger theoretical analysis could likely give much more interesting and applicable results.

When it comes to numerical results, there is no comparison between GAN+SPN and other inference methods. It is then difficult to argue that GAN+SPN could be a promising better method.

**Questions:**

1. I find the "GAN+SPN setup" in equation 4 somewhat confusing and it is hard to fully understand the model.
2. In general, what is GAN+SPN doing exactly and how does this remove the differentiability assumption one would otherwise impose when using a vanilla GAN (or any other generative model)? All this should be explained in detail in a general mathematical model, instead of within a specific physical application.
3. How do we verify the conditions imposed by the theoretical results, so that we know using GAN+SPN is justified?
4. What are the current state-of-the-art inference methods? How do they compare to GAN+SPN?

---

### Official Review · Reviewer_THsC · 2024-11-04

**Soundness:** 2
**Presentation:** 2
**Contribution:** 1
**Rating:** 1
**Confidence:** 4

**Summary:**

The authors introduce a GAN-based framework for estimating simulation parameters from observations. Non differentiable simulation-based steps in the generator are made possible by introducing a score estimation network that predicts discriminator output.

**Strengths:**

The structure and technical explanation of what was done are mostly clearly understandable. The paper does a reasonable job of explaining relevant physical concepts to an ML audience.

**Weaknesses:**

Overall, I find the major problem is that state of the art baselines are not considered, and the tasks are too easy. It's not clear that naive application of standard methods would be worse.

The algorithm should be clearly explained in terms of common deep learning and SBI terminology, separate from the physics application.

There is no comparison to state of the art methods here -- BOLFI and SMC-ABC didn't qualify for that in 2019, and certainly don't now. There's a rich literature on SBI that should be examined, and the leading methods such as SNPE, SNRE, SNLE and their variants should be considered (Lueckmann 2021).

The argument that the proposed method is an advance or not directly comparable to standard sBI methods because it doesn't use parameter priors is fairly weak -- flat (uniform) prior have been used in many SBI studies. While in those cases reparameterization of parameter space could influence the result, the same may well be true for the method introduced here.

From the perspective of the number of parameters estimated and the complexity of the observations, the tasks addressed here fall on the "easy" side of the range of problems addressed with SBI.

GANs are noted for their training difficulties due to using two networks, and the authors propose to add a third. Given that only small networks were used on very small problems, how would this method potentially scale to larger and more challenging tasks?

The key idea of the score estimating network seems to be that we can use the detector simulation in the forward pass. But why not just use a fully differentiable generator to learn the distribution of parameters conditioned on observables in a standard posterior estimation approach to SBI, or observables conditioned on parameters for subsequent MCMC? These are very well-established methods and should be compared (Lueckmann 2021, Tejero-Cantero 2020). Presumably there is some benefit from using this hybrid approach, but we don't see it clearly demonstrated.

Also, the score estimating network seems to have been introduced to differentiate through the detector simulation, but then aspect this is dropped on line 183?

The basics of the algorithm could be better explained in section 2. The notation in eq. 2-3 is nonstandard.

By training a GAN, we are essentially looking for a distribution on $\theta$ that gives the correct marginal on observables $x$. But in many cases there may be infinitely many such distributions, so without a prior isn't the problem severely underdetermined? This would make the size/shape of the recovered distributions on $\theta$ somehwat meaningless. If not, please explain. These concerns are somewhat addressed at the end of section 4, but the argument that the desired parameter distributions are merely a "means to an end" is somewhat unconvincing if the proposed methods are to be of general interest.

I couldn't find details of the QCD task dataset, how many samples were available, how were they generated, etc.

For GAN-based parameter estimation one should also cite and discuss Ramesh (2020) and Tran (2017).

References:

Tran D, Ranganath R, Blei D. Hierarchical implicit models and likelihood-free variational inference. Advances in Neural Information Processing Systems. 2017;30.

Lueckmann JM, Boelts J, Greenberg D, Goncalves P, Macke J. Benchmarking simulation-based inference. InInternational conference on artificial intelligence and statistics 2021 Mar 18 (pp. 343-351). PMLR.

Tejero-Cantero A, Boelts J, Deistler M, Lueckmann JM, Durkan C, Gonçalves PJ, Greenberg DS, Macke JH. SBI--A toolkit for simulation-based inference. arXiv preprint arXiv:2007.09114. 2020 Jul 17.

Ramesh P, Lueckmann JM, Boelts J, Tejero-Cantero Á, Greenberg DS, Gonçalves PJ, Macke JH. GATSBI: Generative adversarial training for simulation-based inference. arXiv preprint arXiv:2203.06481. 2022 Mar 12.

**Questions:**

Despite reading the manuscript as closely as I could, I was unable to understand what the score estimation network is doing, and how the discriminator is related to the score $d \log p(x)/dx$ of the data distribution. Or does "score" here simply mean the logit of discriminator output?

---

### Note · Authors · 2024-11-25

I have read and agree with the venue's withdrawal policy on behalf of myself and my co-authors.